# Biodegradable 3D Printed Scaffolds of Modified Poly (Trimethylene Carbonate) Composite Materials with Poly (L-Lactic Acid) and Hydroxyapatite for Bone Regeneration

**DOI:** 10.3390/nano11123215

**Published:** 2021-11-26

**Authors:** Honglei Kang, Xudong Jiang, Zhiwei Liu, Fan Liu, Guoping Yan, Feng Li

**Affiliations:** 1School of Materials Science and Engineering, Wuhan Institute of Technology, Wuhan 430205, China; kanghonglei@hust.edu.cn (H.K.); polymers2012@163.com (X.J.); polymers20121016@sohu.com (Z.L.); 2Department of Orthopaedics, Tongji Medical College, Huazhong University of Science and Technology, Wuhan 430022, China; lifengmd@hust.edu.cn

**Keywords:** poly (trimethylene carbonate), poly (L-lactic acid), hydroxyapatite, biodegradability, cell proliferation

## Abstract

Biodegradable scaffolds based on biomedical polymeric materials have attracted wide interest in bone transplantation for clinical treatment for bone defects without a second operation. The composite materials of poly(trimethylene carbonate), poly(L-lactic acid), and hydroxyapatite (PTMC/PLA/HA and PTMC/HA) were prepared by the modification and blending of PTMC with PLA and HA, respectively. The PTMC/PLA/HA and PTMC/HA scaffolds were further prepared by additive manufacturing using the biological 3D printing method using the PTMC/PLA/HA and PTMC/HA composite materials, respectively. These scaffolds were also characterized by Fourier transform infrared spectroscopy (FT-IR), gel permeation chromatography (GPC), automatic contact-angle, scanning electronic micrographs (SEM), diffraction of X-rays (XRD), differential scanning calorimetry (DSC), and thermogravimetry (TG). Subsequently, their properties, such as mechanical, biodegradation, cell cytotoxicity, cell compatibility in vitro, and proliferation/differentiation assay in vivo, were also investigated. Experiment results indicated that PTMC/PLA/HA and PTMC/HA scaffolds possessed low toxicity, good biodegradability, and good biocompatibility and then enhanced the cell multiplication ability of osteoblast cells (MC3T3-E1). Moreover, PTMC/PLA/HA and PTMC/HA scaffolds enhanced the adhesion and proliferation of MC3T3-E1 cells and enabled the bone cell proliferation and induction of bone tissue formation. Therefore, these composite materials can be used as potential biomaterials for bone repatriation and tissue engineering.

## 1. Introduction

Bone transplantation is often used in clinical treatment of bone defects. Currently the world-wide demand for bone transplantation has increased significantly [1,2,3]. Autogenous bone transplantation is currently adopted in surgical procedures; however, there are some disadvantages, such as lack of source, difficulty in repairing large areas of bone defect, and some postoperative complications [4,5,6]. Therefore, the development of novel biodegradable scaffolds based on biomedical polymeric materials for bone transplantation has become a current research hot spot [7,8,9].

Benefiting from the porous and network structure provided by the biodegradable polymeric scaffolds, the cells in bone marrow can attach onto them and repair the bone defect without a second surgery. It was reported that biodegradable aliphatic polyesters including the poly(ε-caprolactone) (PCL), poly(L-lactic acid) (PLA), and poly(trimethyl carbonate) (PTMC) revealed a good biological ability for bone defect reparation and drug delivery [10,11,12,13,14]. PTMC is a kind of biomedical polymer, which possesses low toxicity, good biocompatibility, good biodegradability, good flexibility, and good surface erosion properties, and will not produce strong acid compounds after degradation. In addition, PTMC appears as an elastic polymer at body temperature, so it is a good prospect for tissue engineering, controlled drug release, and other biomedical fields. Moreover, PTMC is one of the preferred synthetic thermoplastic polymeric biomaterials that can be used in 3D printing and is suitable for manufacturing the biodegradable scaffold and biofabrication for tissue regeneration and tissue engineering. However, the poor mechanical properties of PTMC generally hampered its widespread application. To address this limitation, PTMC was modified to prepare the composite materials using the copolymerization and blending methods by some polymers and inorganic nanomaterials [15,16,17].

Poly(L-lactic acid) (PLA) is also a biodegradable synthetic polymer and possesses some advantages, such as low toxicity, good biocompatibility, good biodegradability, and good mechanical and physical properties. It can be easily processed in many ways, such as 3D printing, extrusion, spinning, biaxial drawing, and injection blow molding. However, PLA is hard and crisp, with bulk degradability and acidic degradation products, which induces local pH reduction and causes some adverse inflammatory reaction. Therefore, some modification methods are used to remarkably improve its properties including hydrophilicity, degradation, and flexibility [18,19,20]. It is interesting that the properties of PLA and PTMC present good complementarity and that PLA is a good choice to modify PTMC.

Polymer composites with nanoceramics reflected higher mechanical capabilities than pure polymers due to their distribution of nanoparticles within the polymer matrix, which provide adequate strength and stiffness. Before the new tissue repaired the implant, which is gradually degrading, the scaffolds could sustain a certain grade of physical stress [21,22]. Hydroxyapatite (HA) is a natural mineralization form of calcium apatite, which is the crucial component of teeth and the skeleton. Due to its excellent biological capability, low toxicity, and convenience to produce, HA is widely applied in bone and teeth implant operations. Moreover, the HA modified materials such as PLA could enhance its physical and biological properties by suppressing the self-acceleration of acidolysis. In addition, because of the alkalescent HA nanoparticles, the degradation process of ester bonds is also decreased [23,24,25].

The composite materials of PTMC, PLA, and hydroxyapatite (PTMC/PLA/HA) were prepared by the modification and blending of PTMC with PLA and HA. Subsequently, the biodegradable PTMC/PLA/HA scaffolds were further produced by the additive manufacturing of PTMC/PLA/HA composite materials using the biological 3D printing method. Their properties in vitro and in vivo were also evaluated herein. Future research is expected to study their potential applications in bone repair and replantation.

## 2. Materials and Methods

### 2.1. Materials

All chemicals and solvents were of analytical grade. Tin (II) 2-ethylhexanoate (Sn(Oct)_2_) was purchased from Sigma-Aldrich (Louis, MO, USA) and purified by redistillation in vacuo before use. Toluene and tetrahydrofuran (THF) were purified by redistillation over sodium. Triethylamine was refluxed under phthalic anhydride and dried over calcium hydride (CaH_2_) before use. The growth medium was Roswell Park Memorial Institute medium (α-MEM media: 10% fetal bovine serum (Gibco. Co., Waltham, MA, USA), 100 units/mL penicillium, 100 µg/mL streptomycin). PTMC and PLA were synthesized by ring-opening bulk polymerization of trimethylene carbonate (TMC) and L-lactide, respectively, using tin (II) 2-ethylhexanoate as a catalyst [26,27,28]. PTMC was characterized by ^1^H NMR (Varian, Inc. Corporate, Palo Alto, CA, USA), DSC (NETZSCH DSC 200 F3, Erich NETZSCH GmbH & Co. Holding KG, Gebrüder-Netzsch-Strasse, Selb, Germany), automatic contact-angle measurement (SL200A/B/D Series, Solon Tech. Inc. Ltd., Shanghai, China), and gel permeation chromatography (GPC, Waters Corporation Milford, MA, USA). PTMC: ^1^H NMR (300 MHz, CDCl_3_, δ, ppm): 3885 (m, 4H, -COO-CH_2_-CH_2_-CH_2_-), 2035 (m, 2H, -CH_2_-CH_2_-CH_2_-). Glass transition temperature (Tg) of PTMC is −16.7 °C. The molecular weight (Mn) was 3.1 × 10^5^ and polydispersity was 1.45. PLA: ^1^H NMR (300 MHz, CDCl_3_, δ, ppm): 5.2 (m, 2H, C-CH-C), 1.5 (d, 6H, CH_3_). Glass transition temperature (Tg) of PLA is 56 °C. The molecular weight (Mn) was 1.9 × 10^5^ and polydispersity was 1.52. The osteoblast cells MC3T3-E1 were provided by the China Center for Type Culture Collection of Wuhan University, China, and were raised according to the method described in the literature [29].

### 2.2. Preparation of Poly(Trimethylene Carbonate), Poly(L-Lactic Acid) and Hydroxyapatite (PTMC/PLA/HA) Scaffolds

The PTMC/PLA/HA and PTMC/HA composite materials were prepared by the blending of PTMC and HA or PLA. PTMC, PLA and HA (20 nm, purchased from Beijing Deke Daojin Science and Technology Co., China) were added to dichloromethane and stirred to mix them evenly. Under a reduced pressure, the mixture was slowly evaporated to dry and then sheared into small pieces; subsequently, they were dried in condition of vacuum for 48 h to yield the PTMC/PLA/HA composite materials (including PTMC/PLA/0%HA or PTMC/PLA, PTMC/PLA/5%HA, PTMC/PLA/10%HA, PTMC/PLA/15%HA, PTMC/PLA/20%HA, and PTMC/PLA/25%HA containing the HA content percentage to the total polymer quality of 0%, 5%, 10%, 15%, 20%, and 25%, respectively) and PTMC/HA composite materials (including PTMC/0%HA or PTMC, PTMC/5%HA, PTMC/10%HA, PTMC/15%HA, PTMC/20%HA, and PTMC/25%HA containing the HA content percentage to the total polymer quality of 0%, 5%, 10%, 15%, 20% and 25%, respectively).

The 3D printed PTMC/PLA/HA and PTMC/HA scaffolds (diameter × height: 4 mm × 6 mm, 10 mm × 2 mm) were prepared using a Regenovo 3D Bio-Architect^®^ Pro Biological 3D printer (Hangzhou Regenovo Biotechnology, Ltd., Hnagzhou, Zhejiang, China) with the pieces of PTMC/PLA/HA and PTMC/HA composites, respectively. Inner diameter of print head was ≥0.41 mm, extrusion pressure of print head was 0.3 MPa, and printing rate was 6 mm/s at room temperature.

### 2.3. In Vitro Degradation Test

The PTMC, PTMC/HA, and PTMC/PLA/HA scaffolds (0.1 g, diameter × height: 10 mm × 2 mm) were suspended in 10 mL of PBS (pH 7.4) in a sealed dialysis bag and then 90 mL of PBS was added and it was shaken in a 250 mL Erlenmeyer flask at 37 °C. The samples were taken out of the degradation medium at 1, 2, 3, 4, 5, and 6 months, respectively. They were then rinsed with distilled water and then dried for 48 h in vacuo. The molecular weight and weight loss of the scaffold at different degradation times were calculated.

### 2.4. In Vitro Drug Release Study

5-DOX-incorporated PTMC, PTMC/HA, or PTMC/PLA/HA composite materials were obtained by blending doxorubicin (DOX, 10 mg) and PTMC, PTMC/HA, or PTMC/PLA/HA composite materials (100 mg) in 20 mL THF and drying under vacuum. Next, 5-DOX-incorporated PTMC, PTMC/HA, or PTMC/PLA/HA scaffolds were prepared using the according composites in a disc mold (diameter × height: 10 mm × 2 mm) on a thermocompressor with loading pressure of 10 MPa, setting temperature of 110 °C, and molding time of 10 min.

The 5-DOX-incorporated PTMC, PTMC/HA, or PTMC/PLA/HA scaffolds were suspended in 10 mL of PBS in a sealed dialysis bag and then shaken in 90 mL of PBS at 37 °C in a 250 mL Erlenmeyer flask. A part of the solution outside the dialysis membrane were replaced with same volume of PBS at various times intervals and tested at 256 nm by UV-Vis spectrophotometer (UV-2800 series, Unico, Shanghai, China). The changes of the concentrations of DOX were obtained from curves of the absorption A versus concentration C of DOX in PBS based on the Lambert-Beer law.

### 2.5. Cell Cytotoxicity Assay

For the cell cytotoxicity assay, the osteoblast cells MC3T3-E1 were provided by the China Center for Type Culture Collection of Wuhan University, China and kit 8 (CCK-8) assay was performed according to the manufacturer’s protocol. Briefly, a series of PTMC, PTMC/HA, and PTMC/PLA/HA scaffolds were produced by 3D printing, the parameters were set as follows: thickness = 2 mm, diameter = 6 mm, and the content percentage of HA to the total polymer quality was set as 0%, 5%, 10%, 15%, 20%, and 25%. All scaffolds were rinsed in the distilled water and dried, and then they were sterilized by using ethylene oxide. MC3T3-E1 cells were counted and seeded on the surface of scaffolds in 48-well culture plates at a density of 2 × 10^4^ cells/well; cells were then cultured with complete α-MEM medium for one and three days; subsequently, a CCK-8 assay was conducted at each time point and tested at a wave length of 450 nm.

### 2.6. Cell Attachment and Proliferation Assay

For the cell attachment assay, two types of scaffolds formed with PTMC/HA and PTMC/PLA/HA composites containing the different HA content, which were made by 3D printing, were produced with a diameter of 6 mm and athickness of 2 mm. The scaffolds were precleaned and ethylene oxide sterilization was performed, MC3T3-E1 cells were then seeded onto the scaffolds in 24-well culture plates at a density of 1 × 10^5^ cells/well and were cultured with α-MEM medium for 7 days. Next, rhodamine labeled phalloidin staining was performed. Briefly, cells were washed with PBS one time and fixed with 4% paraformaldehyde for 15 min; after washing 3 times with PBS, phalloidin staining assay was added and incubated for 1 h, after then 4′,6-diamidino-2-phenylindole (DAPI, Invitrogen, USA) staining was conducted for 15 min. Subsequently, cells were washed 3 times and images were captured with EVOS fluorescence microscope (Thermo Fisher Scientific, USA). For proliferation assay, cells were treated with 3-(4,5-dimethyl-2-thiazolyl)-2,5-diphenyl-2-H-tetrazolium bromide (MTT) assay at 1 and 2 days according to the manufacturer’s specification and tested at wavelength of 570 nm.

### 2.7. Osteogenic Gene Expression

For osteogenic related gene expression analysis, real time-quantitative polymerase chain reaction (RT-qPCR) was performed. Briefly two types of scaffolds were produced and sterilized before using, and the MC3T3-E1 cells were provided by the China Center for Type Culture Collection of Wuhan University, China. They were seeded onto scaffolds in 6-well culture plates at a density of 8 × 10^6^ cells per well. MC3T3-E1 cells were incubated with osteogenic differentiation medium (α-MEM medium, 100 U/mL penicillin, 100 mg/mL streptomycin, 10 nmol/L dexamethasone, 10 mmol/L β-glycerolphosphate, 10 nmol/L vitamin D, and 10% fetal bovine serum) for 7 days, then total RNA was obtained by using Trizol reagent (Invitrogen) according to the standard protocol. Subsequent 1 µg total RNA was used for reverse transcription in to cDNA with Reverse Transcription Kit (TOYOBO, Kita-ku, Osaka 530-8230, Japan), Finally, qPCR analysis was conducted by SYBR assay (KAPA, Boston, MA, United States), and relative expression of osteogenesis related genes were analyzed by the 2–ΔΔCT method. The mRNA expression level was normalized to the reference gene β-Actin. The specific primers used for PCR amplification were as follows (F, forward; R, reverse):

β-Actin, F 5′-GTGACGTTGACATCCGTAAAGA -3′ and R 5′-GCCGGACTCATCGT ACTCC-3′, Osteocalcin (OCN) F 5′-CTGACCTCACAGATCCCAAGC and R 5′-TGGTCTGATAGCTCGTCACAAG, Runt-related transcription factor 2 (RUNX2) F 5′-GACTGTGGTTACCGTCATGGC and R 5′-ACTTGGTTTTTCATAACAGCGGA, A Lkaline Phosphatase (ALP) F 5′-GCCTTACCAACTCTTTTGTGCC and R 5′-CACCCGAGTGGTAGTCACAAT, Type I collagen (Col I )F 5′-TAAGGGTCCCCAATGGTGAGA and R 5′-GGGTCCCTCGACTCCTACAT.

### 2.8. In Vivo Implanted Assay of PTMC/HA and PTMC/PLA/HA Scaffolds in Femur Defect

Sprague Dawley (SD) rats (weighted approximately 450g) were obtained from the Experimental Animal Center of Tongji Medical College (Wuhan, China). All rats were fed in the Specific Pathogen Free (SPF) room with professional equipment and abundant sterile water and food. They were randomly separated into three groups (six rats per group): control, PTMC/HA group, and PTMC/PLA/HA group. Two types of scaffolds were produced by 3D printing with parameters as follows: HA content = 25%, diameter = 4 mm, and height = 6 mm. All were rinsed and sterilized with ethylene oxide before the surgery. SD rats were anesthetized with 3% pentobarbital, then the femur was exposed and a 4 mm defect was made at the external epicondyle of the femur. The muscle was used to press the defect hole in the control group; for the two scaffolds groups, PTMC/PLA/25%HA and PTMC/25%HA scaffolds were implanted into it, respectively. After the operation, all rats were injected with penicillin solution for 3 days and fed for another 2 months. Subsequently, the rats were euthanized, and the femurs were fixed with 4% paraformaldehyde. For evaluation of the defect reparation and new bone formation in the scaffolds, micro-computed tomography (μ-CT, Scanco Medical) scanning was performed: the scanning parameters were set as 100 kV, 98 μA, and voxel size = 10 μm. Fracture surface and three-dimensional images were obtained by using software built in the Micro CT Scanner, the cylindrical area based on the defect area was calculated for new bone formation, and the main parameters including trabecular numbers (Tb.N), trabecular space (Tb.Sp), bone volume/tissue volume (BV/TV), and trabecular thickness (Tb.Th) were analyzed.

### 2.9. Statistical Analysis

For statistical analysis, all experiments were conducted independently ≥3 times. Images represent typical results of experiments, and the results were revealed as the means ± SD. T-test methods were used to analyze differences between two groups, and one-way ANOVA was performed among more than two groups. A *p* value of less than 0.05 was regarded as a significant difference.

## 3. Results and Discussion

### 3.1. Characterization of PTMC/HA and PTMC/PLA/HA Scaffolds

The biodegradable PTMC/HA and PTMC/PLA/HA scaffolds were produced by the blending and further 3D printing of composite materials based on PTMC, PLA, and HA. In the PTMC/PLA/HA scaffolds, PTMC was chosen as a soft polymer segment, PLA was used as a rigid polymer segment, and HA was used as a reinforcing filler. PTMC/PLA/HA composite materials were prepared by the modification of PTMC by PLA and HA and were expected to improve the mechanical strength and mechanical properties while maintaining flexibility. Moreover, the alkalescent HA nanoparticles can neutralize the acidic matrix of PLA and suppress the degradation process of ester bonds further.

The PTMC/HA and PTMC/PLA/HA scaffolds were characterized by FT-IR, automatic contact-angle, DSC, TG, XRD, SEM, and mechanical properties. As shown in Figure 1, the chemical characterization of pure HA, PTMC, and the two types of scaffolds printed by PTMC/HA and PTMC/PLA/HA composites (containing various content of HA) were performed using FT-IR analysis. When these substrates were mixed together, many of the absorption bands overlapped and the capability of the functional groups was retained. However, the location, shape, and peak strengths changed greatly. The spectra of PTMC/HA scaffolds reflected a typical CH peak (2923 cm^−1^) and ester peaks (respectively at 1748, 1175, 1030 cm^−1^), and were accompanied by increased HA content percentage in scaffolds ranging from 0% to 25%, with a corresponding decrease in peak intensities [30]. Moreover, PTMC/HA scaffolds revealed spectral features at 1080, 960, 603, and 564 cm^−1^, which reflected the vibrations of *p* = O and *p*-O in HA, and ranged from 3500 to 3200 cm^−1^, which indicated the OH groups’ absorption peak in the HA and PTMC. However, their peak intensities decreased greatly (Figure 1a). The spectra of PTMC/PLA/HA scaffolds showed typical CH peaks (2963 cm^−1^) and ester peaks (at 1948, 1181, 1060 cm^−1^), and were accompanied by increased HA content percentage in scaffolds ranging from 0% to 25%; the peak intensities were also decreased. Moreover, PTMC/PLA/HA scaffolds showed typical spectral features at 1080, 960, 615, and 564 cm^−1^, which showed the vibrations of *p* = O and *p*-O in HA, and ranged from 3500 to 3200 cm^−1^, which revealed the OH groups’ absorption peak in the HA and PTMC, and their peak intensities declined greatly (Figure 1b).

Water contact angles of PTMC/HA and PTMC/PLA/HA scaffolds are shown in Figure 1. Water contact angles of PTMC/HA decreased along with the increase in HA content from 0 to 20% and then increased when HA content percentage continuously increased from 20% to 25% (Figure 1b). The water contact angle of PTMC/HA scaffolds with 20% HA content displayed the lowest water contact angles of 84.46°. Water contact angles of PTMC/PLA/HA decreased along with the increase in HA content from 0 to 15% and then increased when HA content percentage continuously increased from 15% to 25% (Figure 1d). The water contact angle of PTMC/PLA/HA scaffolds with 15% HA content displayed the lowest water contact angles of 56.80°. Moreover, PTMC/PLA/HA scaffolds possessed lower water contact angles and higher hydrophilicity than the PTMC/HA scaffolds. Therefore, the modification of PLA and HA to PTMC can enhance the hydrophilicity of composite materials.

Thermal properties of PTMC/HA and PTMC/PLA/HA scaffolds were measured by DSC and TG (Figure 2a–d and Table 1 and Table 2). Figure 2 indicate the DSC curves of PTMC, PTMC/HA, and PTMC/PLA/HA composite materials. PTMC/HA indicated the typical peaks of glass-transition temperature (Tg) ranging from −18 °C to −16 °C and Tg of pure PTMC was −16.7 °C, indicating HA had good compatibility with PTMC. It can be seen from Figure 1c that the temperature of the melting peak of PTMC was shifted by the addition of HA, which indicates that HA has some influence on the crystalline integrity and grain size of PTMC materials. The increase in HA content caused damage to PTMC crystal particles a decrease in crystal confinement and crystalline integrity. In contrast, the hydrogen-bonding interaction of hydroxyl groups on the surface of HA with PTMC increased. Therefore, the melting peak temperature of PTMC/HA composites decreased gradually along with the increase in HA content from 0 to 10%, reaching the minimum when the HA content was 10%. Subsequently, the melting peak temperature of the composites gradually increased again while the HA content continued to increase from 10% to 25%.

However, PTMC/PLA/HA revealed two typical Tg related peaks that ranged from −14 °C to −12 °C and varied from 46 °C to 56 °C when Tg of pure PTMC and PLA were −16.7 °C and 56 °C, respectively, indicating both PTMC and PLA existed in the composite materials. It can be seen from Figure 1d that the temperature of the melting peak of PTMC was highly shifted and the temperature of the melting peak of PLA was lowered by the addition of HA, which indicated that HA has some influence on the crystalline integrity and grain size of PTMC and PLA materials. An increase in HA content caused damage to the PTMC and PLA crystal particles and a decrease in crystal confinement and crystalline integrity. In contrast, the hydrogen-bonding interaction of hydroxyl groups on the surface of HA with PTMC and PLA increased. Therefore, the melting peak temperature of PTMC component (T_g1_) in PTMC/PLA/HA composites increased gradually along with the increased HA content percentage ranging from 0% to 15%, reaching the maximum when the HA content was 15%. Subsequently, the melting peak temperature of PTMC component in the composites gradually decreased again while the HA content continued to increase from 15% to 25%. However, the melting peak temperature of PLA component (T_g2_) in PTMC/PLA/HA composites decreased gradually along with the increased HA content ranging from 0% to 15%, reaching the minimum when the HA content was 15%. Subsequently, the melting peak temperature of PLA component in the composites gradually increased again while the HA content continued to increase from 15% to 25%. Moreover, the difference value of melting peak temperature (T_g2_-T_g1_) of PTMC/PLA/HA composites decreased gradually along with the increased HA content, reaching the minimum when the HA content was 15%. Subsequently, the difference value of melting peak temperature (T_g2_-T_g1_) of the composites gradually increased again while the HA content continued to increase from 15% to 25%. These results demonstrate the addition of HA can promote compatibility between PTMC and PLA.

Figure 2c,d and Table 2 show TG of pure PTMC, PTMC/HA, and PTMC/PLA/HA composite materials, respectively, and then represent thermal degradation of PTMC, PTMC/HA, and PTMC/PLA/HA. The effect of HA contents on the thermal properties of PTMC/HA and PTMC/PLA/HA scaffolds were analyzed. The thermal stability of PTMC was better than that of PTMC/HA composite materials. PTMC began to degrade at 258.6 °C and stopped at 307.9 °C, which indicated that PTMC had high thermal stability. For PTMC/HA composites, the addition of HA slightly reduced the thermal stability of the composites, and the thermal degradation curve moved to the low temperature zone. When the HA content increased from 0% to 25%, the degradation temperature for weight loss at 10% and 50% (T_10%_ and T_50%_) showed a trend of decreasing gradually, and both degradation temperature for weight loss at 80% and maximum value (T_80%_ and T_max_) were increasing.

The thermal stability of PTMC/PLA/HA composites with lower HA content were better than that of PTMC/PLA composites; however, thermal stability of PTMC/PLA/HA composites decreased gradually with the increase in HA content. PTMC/PLA began to degrade at 279.4 °C and stopped at 386.3 °C, which indicated that PTMC/PLA had higher thermal stability than that of pure PTMC and PTMC/PLA/HA had higher thermal stability than that of PTMC/HA. For PTMC/PLA/HA composites, the addition of HA slightly decreased the composites’ thermal stability, and the thermal degradation curve moved to the low temperature zone.

The degradation temperature accompanied by a weight loss at 10%, 50%, 80%, and the maximum (T_10%_, T_50%_, T_80%_ and T_max_) of PTMC/PLA/HA composites increased gradually along with the increased content of HA ranging from 0% to 15%, reaching the maximum when the content of HA was 15%. Subsequently, the degradation temperature with a weight loss of 10%, 50%, 80%, and the maximum (T_10%_, T_50%_, T_80%_ and T_max_) of PTMC/PLA/HA composites gradually decreased again while the HA content continued to increase from 15% to 25%. These results also showed the addition of HA can promote the compatibility between PTMC and PLA.

### 3.2. Scanning Electronic Micrographs of PTMC/HA and PTMC/PLA/HA Scaffolds

The micrographs of PTMC/HA and PTMC/PLA/HA scaffolds were examined by SEM. Figure 3 shows that all PTMC/HA and PTMC/PLA/HA scaffolds presented random morphology. HA nanoparticles scattered uniformly in the scaffolds and there were a few aggregated nanoparticles. The intensity of HA nanoparticles appeared to increase according to the enlargement of HA content in scaffolds. The micrographs of PTMC/HA scaffolds showed that HA had good compatibility with PTMC. However, some hierarchical structures appear in the micrographs of PTMC/PLA/HA scaffolds, which indicates that both PTMC and PLA existed in the composite materials and the addition of HA can promote the compatibility between PTMC and PLA. These results are consistent with that of the above thermal properties of PTMC/HA and PTMC/PLA/HA scaffolds. Typical XRD spectra obtained for PTMC/HA and PTMC/PLA/HA scaffolds are presented in Figure 3d1,d2. The spectra of PTMC/HA and PTMC/PLA/HA scaffolds also showed the same characteristic peaks as PTMC. Three-dimensional printed PTMC/HA and PTMC/PLA/HA scaffolds possessed different uniform pore size structures.

The compressive strength and modulus of PTMC/HA and PTMC/PLA/HA scaffolds were measured and are shown in Figure 4a and Table 3b. The compressive strength and modulus of PTMC/HA scaffolds increased along with the increased HA content ranging from 0% to 25%. HA was uniformly dispersed in the PTMC polymer matrix. When the composite was affected by external force, it blocked the movement of the molecular chain, thus increasing the compressive properties of the PTMC/HA composites. Moreover, PTMC/HA and PTMC/PLA/HA scaffolds possessed the obviously higher compressive strength and modulus than that of PTMC and PTMC/HA scaffolds, respectively. These results showed that the addition of HA can promote the compressive property of PTMC/HA. The compressive strength and modulus of PTMC/PLA/HA scaffolds increased along with the increase in HA content from 0 to 5%, reaching the maximum value when the HA content was 5%. Subsequently, the compressive strength and modulus of PTMC/PLA/HA composites gradually decreased while the HA content continued to increase from 5% to 25%.

### 3.3. In Vitro Degradation

The in vitro degradation of PTMC/HA and PTMC/PLA/HA scaffolds were measured in PBS at 37 °C. During the degradation process in PBS for 6 months, PTMC, PTMC/25%HA, PTMC/PLA, and PTMC/PLA/HA scaffolds with 10% and 25% HA content (PTMC/PLA/10%HA and PTMC/PLA/25%HA) experienced weight loss, and molecular weight loss increased with time. The weight losses of PTMC/PLA, PTMC/PLA/10%HA, and PTMC/PLA/25%HA scaffolds were 8%, 12%, and 32% at the end of 6 months, respectively (Figure 5b). PTMC/PLA/25%HA and PTMC/25%HA scaffolds accordingly displayed the obviously higher weight loss compared to that of PTMC/PLA and PTMC scaffolds, respectively. The PTMC/PLA/10%HA scaffolds showed the slightly higher weight loss than that of PTMC/PLA and PTMC scaffolds and the lower weight loss than that of PTMC/25%HA and PTMC/PLA/25%HA scaffolds accordingly.

The molecular weight of PTMC/PLA and PTMC/PLA/25%HA scaffolds decreased significantly, whereas that of PTMC/PLA/10%HA scaffolds changed relatively little (Figure 5d, Appendix A). The degradation of PLA resulted in local pH reduction, which accelerated the degradation of the PTMC/PLA and PTMC/PLA/HA composites. The PTMC/PLA composite materials contained relatively more PLA and then PTMC/PLA scaffolds, indicating high molecular weight loss after 6 months degradation. The PTMC/PLA/25%HA scaffolds contained the largest amount of HA, which perhaps aggregated as nanoparticles and formed the large holes and cracks in the scaffolds after the removal of HA from the composites in the degradation process. In addition, the increase in water permeation further accelerated hydrolysis and degradation. Thus PTMC/PLA/25%HA scaffolds had the highest molecular weight and also weight loss due to the high permeability and promoted degradation rate. The PTMC/PLA/10%HA scaffolds had lower weight loss, molecular weight loss, and degradation rate than PTMC/PLA/25%HA scaffolds because HA nanoparticles scattered uniformly in PTMC/PLA composite matrix. Because of the alkalescent HA nanoparticles, it could the neutralize acidic substances in PLA and further suppress the degradation process of ester bonds. These results indicated that hydroxyl groups of HA and pores in the matrix enhanced the hydrophilicity, water absorption, and degradation rate of scaffolds.

### 3.4. In Vitro Drug−Release Properties of PTMC/HA and PTMC/PLA/HA Scaffolds

DOX was used as a typical model anticancer drug model for the drug-release properties of PTMC/HA and PTMC/PLA/HA scaffolds because it is easy to detect by HPLC and UV. The DOX release properties of pure PTMC, PTMC/HA, and PTMC/PLA/HA scaffolds are shown in Figure 6. The substantial releases of the DOX-incorporated pure PTMC, PTMC/HA, and PTMC/PLA/HA scaffolds were maintained for 65 days of measurement. The DOX-incorporated pure PTMC, PTMC/HA, and PTMC/PLA/HA scaffolds displayed steady drug-release rates and good drug-controlled release properties.

Compared with DOX-incorporated pure PTMC scaffolds, DOX-incorporated PTMC/HA and PTMC/PLA/HA scaffolds showed faster drug-release rates, presumably due to the increased drug diffusion coefficient of PTMC/HA and PTMC/PLA/HA scaffolds. Moreover, the release rate increased along with an increase in HA content. The cumulative percentage release of PTMC/10%HA and PTMC/25%HA scaffolds reached 10.8% and 11.6%, respectively, and higher than that of pure PTMC scaffolds (8.0%), after 45 days of drug-release. The cumulative percentage release of PTMC/PLA/10%HA and PTMC/PLA/25%HA scaffolds reached 7.3% and 8.1%, respectively, and higher than that of PTMC/PLA scaffolds (6.4%), after 55 days of drug-release. This result indicated that hydroxyl groups of HA and pores in the matrix enhanced the hydrophilicity, water absorption, and drug diffusion coefficients of scaffolds.

### 3.5. Cell Cytotoxicity Assay

Metabolic activity of the osteoblast cells MC3T3-E1 cultured with pure PTMC, PTMC/HA, and PTMC/PLA/HA scaffolds was evaluated by the CCK-8 assay on 1- and 3-T3-E1 cells. There were no significant differences of cell proliferation activity among all scaffolds at 1 day. The cell proliferation on PTMC/15%HA and PTMC/25%HA scaffolds was slightly higher than that of the other PTMC/HA scaffolds. The cell proliferation on PTMC/PLA/5%HA and PTMC/PLA/10%HA scaffolds was slightly higher than that of the other PTMC/PLA/HA scaffolds. Moreover, PTMC/PLA/HA scaffolds possessed lower cell cytotoxicity and higher cell proliferation than that of PTMC/HA scaffolds.

However, these scaffolds revealed a different cell proliferation capacity at 3 days. PTMC/25%HA scaffolds and pure PTMC had the higher cell proliferation activity than that of the other PTMC/HA scaffolds. PTMC/PLA scaffolds displayed lower cell proliferation activity than that of PTMC/PLA/25%HA scaffolds, but higher than that of the other PTMC/PLA/HA scaffolds with 5%, 10%, 15%, and 20% HA content. Moreover, PTMC/PLA/HA scaffolds also possessed lower cell cytotoxicity and higher cell proliferation than that of PTMC/HA scaffolds. These results demonstrate that PTMC/PLA/25%HA and PTMC/25%HA scaffolds indicated good biocompatibility and showed a good biological ability for cell proliferation.

According to the analysis of experimental results of cell cytotoxicity assay, FT-IR, automatic contact-angle, DSC, TG, XRD, SEM, and compressive modulus above, PTMC/25%HA and PTMC/PLA/25%HA possessed the highest cell proliferation performances and were chosen as the samples to further investigate the cell viability in vitro and proliferation and new bone regeneration with femur defect model in vivo. Figure 6 shows the 3D printing composite scaffold with obvious network structure produced by 3D printing technology. Through the test of shrinkage and porosity of the scaffold, 3D printed PTMC/25%HA scaffolds showed porosity of 65 ± 0.82% and shrinkage of 15.6 ± 0.36%, whereas 3D printed PTMC/PLA/25%HA scaffolds displayed porosity of 67 ± 0.82% and shrinkage of 15.8 ± 0.29%.

### 3.6. In Vitro Cell Studies of PTMC/25%HA and PTMC/PLA/25%HA Scaffolds

Metabolic activity of the osteoblast cells MC3T3-E1 cultured with the PTMC/25%HA and PTMC/PLA/25%HA scaffolds were evaluated by the 3-(4,5-dimethyl-2-thiazolyl)-2,5-diphenyl-2-H-tetrazolium bromide (MTT) assay, real time fluorescent quantitative PCR, and Western blot assay at 7 days postculturing. The expression of β-Actin, collagen (COL), Osteocalcin (OCN), Alkaline phosphatase (ALP), and Runt related transcription factor-2 (RUNX2) at mRNA level were performed by using qPCR and are shown in Figure 8c. As showed in Figure 8a–c, PTMC/PLA/25%HA scaffolds displayed higher cell adhesion capacity and proliferation activity than that of PTMC/25%HA scaffolds by the phalloidin staining and MTT assay.

PTMC/25%HA and PTMC/PLA/25%HA scaffolds supported the growth of MC3T3-E1 cells and promoted the expression of COL I, OCN, ALP, and RUNX2 at mRNA level. Moreover, MC3T3-E1 cells scattered uniformly and displayed the good adhesion and cell proliferation activity on the surface of 3D branches in PTMC/25%HA and PTMC/PLA/25%HA scaffolds (Figure 8c). PTMC/25%HA and PTMC/PLA/25%HA scaffolds evidently promoted the expression of COL Ⅰ, OCN, ALP, and RUNX2. These osteogenesis related genes induced by PTMC/25%HA scaffolds indicated a higher expression level than that of PTMC/PLA/25%HA scaffolds. These results demonstrate that both types of scaffolds revealed a good biocompatibility, and PTMC/25%HA scaffolds showed a good capacity at osteogenesis.

### 3.7. PTMC/25%HA and PTMC/PLA/25%HA Scaffolds Promoted the Bone Reparation in Femur Defect Model

Groups of SD rats were given femur defects (diameter = 4 mm, the height = 6 mm) at the external tuberosity of the femur according to the method reported in previous studies [30,31]. Th SD rats of the control groups were pressed with the muscle and sutured; the PTMC/25%HA and PTMC/PLA/25%HA scaffold groups were implanted with the 3D printed and sterile scaffolds. After 2 months, the femurs were fixed with 4% paraformaldehyde and μ-CT scanning was performed; the cylindrical area based on the defect area was calculated for new bone formation. Figure 9a reveals the good biological capacity of PTMC/25%HA and PTMC/PLA/25%HA scaffolds, which indicated among the porosity and the branches, abundant new bone formation appeared and repaired the bone defect. At the same time, the PTMC/25%HA and PTMC/PLA/25%HA scaffolds were slowly biodegraded.

Compared with the PTMC/PLA/25%HA scaffold group, the PTMC/25%HA scaffolds showed more new bone tissue and the data analysis indicated a higher bone tissue volume/total tissue volume (BV/TV), larger trabecular thickness (Tb.Th), higher trabecular space (Tb.Sp) and lower number of trabecular (Tb.N). It is possible that the prolonged acidic byproducts of PLA degradation caused negative effects on the cell proliferation and bone reparation.

## 4. Conclusions

The PTMC/HA and PTMC/PLA/HA scaffolds were prepared by 3D printing of composite materials based on PTMC, PLA, and HA and possessed good biodegradability, good biocompatibility, and improved cell proliferation of MC3T3-E1. Moreover, PTMC/HA and PTMC/PLA/HA scaffolds provided an appropriate microenvironment for osteogenesis and then displayed remarkable osteogenic activity. Therefore, PTMC/HA and PTMC/PLA/HA composite materials enable the proliferation of bone cells in vitro and bone tissue regeneration in vivo; these materials could be potential biomaterials for bone repatriation and tissue engineering. Furthermore, according to the analysis of experimental results, PTMC/25%HA and PTMC/PLA/25%HA scaffolds displayed the best performing biomaterials for bone regeneration. In addition, PTMC/PLA/25%HA scaffolds possessed the better mechanical properties, lower cell cytotoxicity, and higher cell proliferation than that of PTMC/25%HA scaffolds.

## Figures and Tables

**Figure 1 nanomaterials-11-03215-f001:**
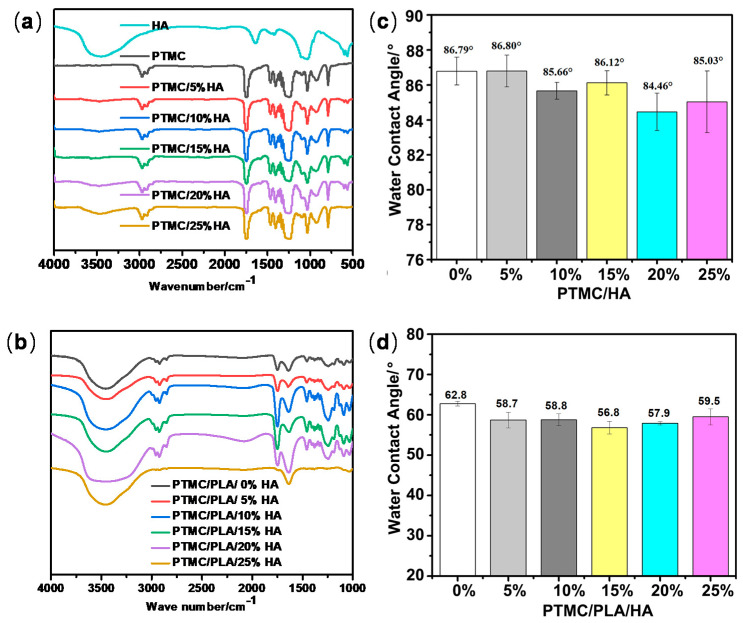
FT-IR (**a**) and water contact angle (**c**) of PTMC/HA; FT-IR (**b**) and water contact angle (**d**) of PTMC/PLA/HA scaffolds.

**Figure 2 nanomaterials-11-03215-f002:**
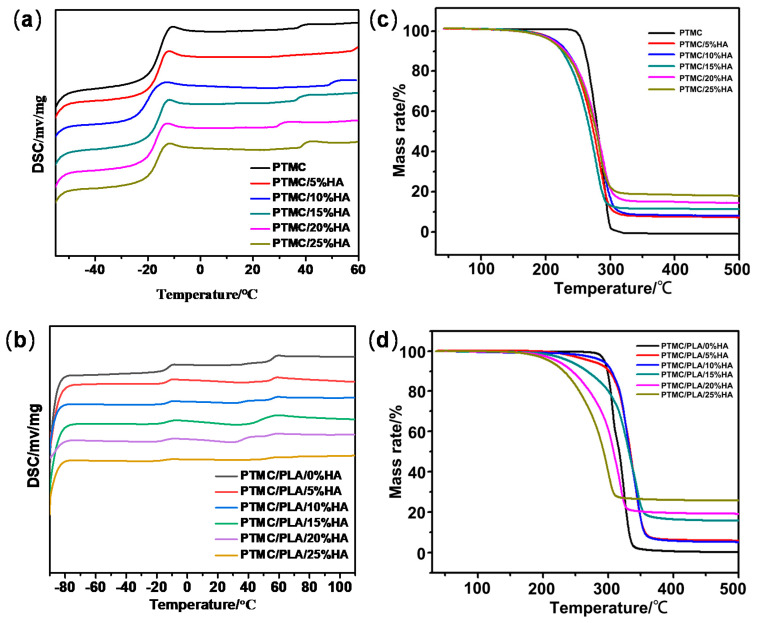
DSC (**a**) and TG (**c**) of PTMC/HA; DSC (**b**) and TG (**d**) of PTMC/PLA/HA scaffolds.

**Figure 3 nanomaterials-11-03215-f003:**
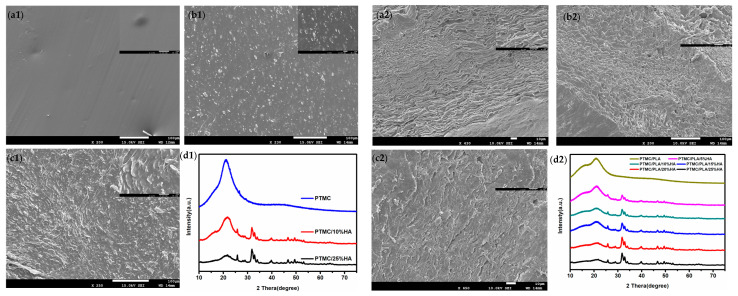
Micromorphology of PTMC/HA and PTMC/PLA/HA scaffold. (**a1**): PTMC, (**b1**): PTMC/10%HA, (**c1**): PTMC/25%HA, (**a2**): PTMC/PLA, (**b2**): PTMC/PLA/10%HA, (**c2**): PTMC/PLA/25%HA); XRD of PTMC/HA (**d1**) and PTMC/PLA/HA scaffolds (**d2**).

**Figure 4 nanomaterials-11-03215-f004:**
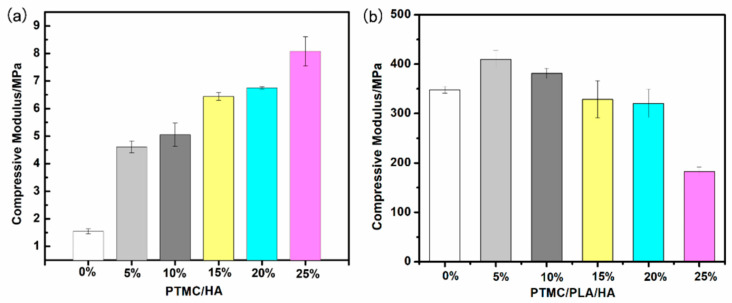
Compressive modulus of PTMC/HA (**a**) and PTMC/PLA/HA (**b**) scaffolds.

**Figure 5 nanomaterials-11-03215-f005:**
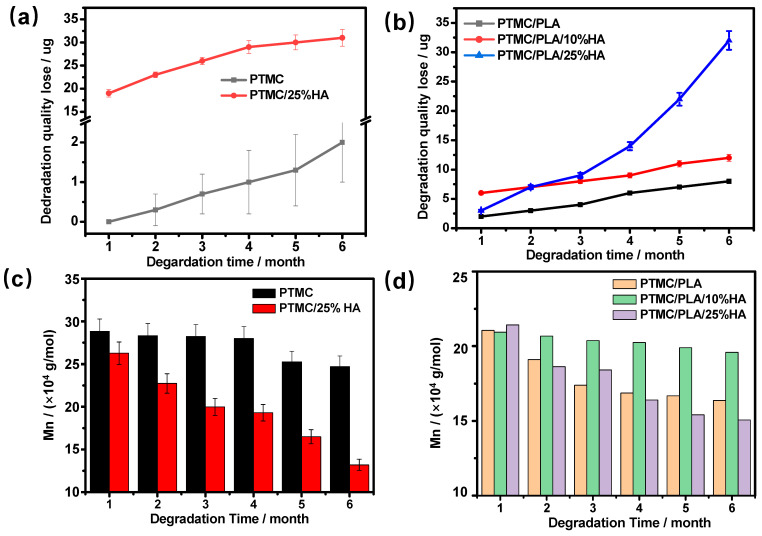
Weight loss (**a**,**b**) and number average molecular weight loss (**c**,**d**) of PTMC/PLA and PTMC/PLA/HA scaffolds.

**Figure 6 nanomaterials-11-03215-f006:**
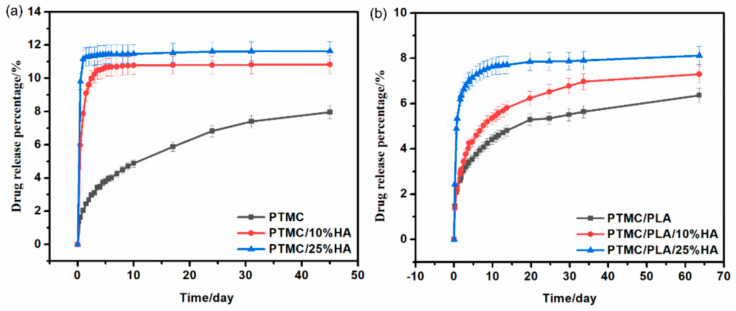
Release profiles of DOX from PTMC/HA (**a**) and PTMC/PLA/HA (**b**) scaffolds.

**Figure 7 nanomaterials-11-03215-f007:**
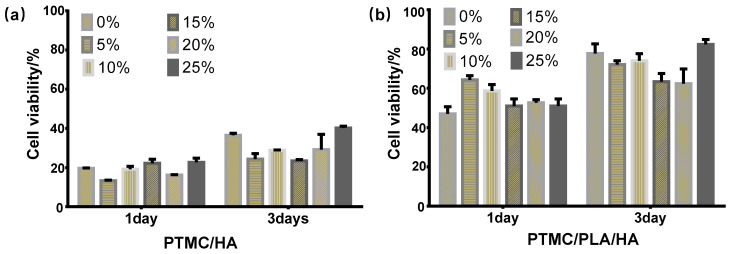
Cell proliferation assays of the osteoblast cells MC3T3-E1. (**a**) PTMC/HA scaffolds containing 0%, 5%, 10%, 15%, 20%, and 25% HA content, (**b**) PTMC/PLA/HA scaffolds containing 0%, 5%, 10%, 15%, 20%, and 25% HA content.

**Figure 8 nanomaterials-11-03215-f008:**
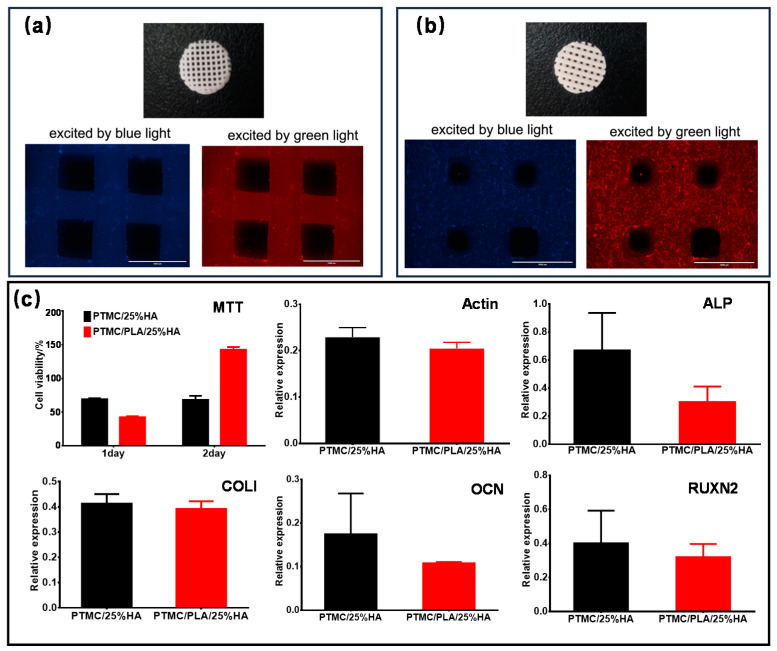
Phalloidin staining images of cells after 7 days culturing on the scaffolds of (**a**) PTMC/25%HA scaffolds, and (**b**) PTMC/PLA/25%HA scaffolds. (**c**) MTT assay and effects of the scaffolds on mRNA expression of the Actin, Runx2, ALP, and Col I genes on day 7.

**Figure 9 nanomaterials-11-03215-f009:**
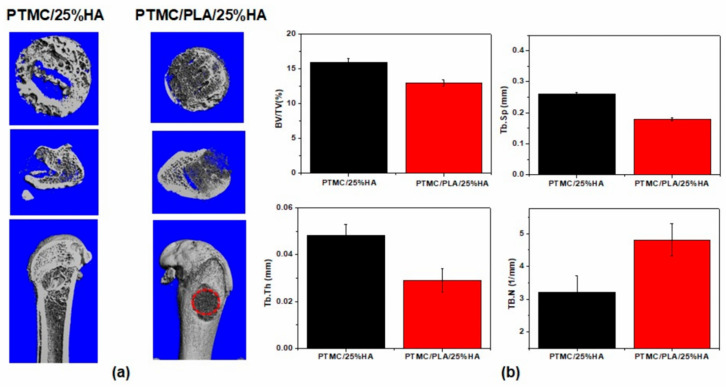
Micro-CT analysis of the effect of scaffolds on femur defect in vivo. (**a**) Representative three-dimensional reconstructed micro-CT images showing the effect of PTMC/25%HA and PTMC/PLA/25%HA scaffolds on the new bone tissue formation inside the defect site (red dashed line). (Left: PTMC/25%HA scaffolds after 2 months, right: PTMC/PLA/25%HA scaffolds after 2 months.) (**b**) Summarized data showing the micro-architectural parameters of the new formed bone tissue at 2 months by analyzing the three-dimensional reconstructed micro-CT images using image analysis software. BMD, BV/TV, Tb.Th, and Tb.N were shown in the panel.

**Table 1 nanomaterials-11-03215-t001:** DSC data of PTMC/HA and PTMC/PLA/HA scaffolds.

	T_g1_ (°C)	T_g2_ (°C)	T_g2_-T_g1_ (°C)
PTMC	−16.7		
PTMC/5%HA	−17.0		
PTMC/10%HA	−17.5		
PTMC/15%HA	−16.9		
PTMC/20%HA	−16.7		
PTMC/25%HA	−16.2		
PTMC/PLA	−13.9	54.9	68.8
PTMC/PLA/5%HA	−12.9	55.6	68.5
PTMC/PLA/10%HA	−13.0	55.0	68.0
PTMC/PLA/15%HA	−12.2	46.6	58.9
PTMC/PLA/20%HA	−13.0	54.9	67.9
PTMC/PLA/25%HA	−15.6	55.0	70.6

**Table 2 nanomaterials-11-03215-t002:** Thermal properties of PTMC/HA and PTMC/PLA/HA scaffolds.

Heating Rate (°C/min)	T_10%_ (°C)	T_50%_ (°C)	T_80%_ (°C)	T_max_ (°C)	Residue at 400 °C (%)	Residue at 450 °C (%)
PTMC	258.6	280.6	291.7	307.9	0.3	0.3
PTMC/5%HA	227.1	274.1	292.5	329.6	8.6	8.3
PTMC/10%HA	230.1	278.4	297.7	331.2	9.3	9.1
PTMC/15%HA	223.2	268.9	287.6	337.2	12.4	12.3
PTMC/20%HA	227.8	280.1	304.1	342.1	15.7	15.4
PTMC/25%HA	224.8	277.9	317.9	349.5	19.2	18.9
PTMC/PLA	298.9	317.0	328.1	376.3	0.7	0.4
PTMC/PLA/5%HA	302.5	335.8	347.1	380.8	6.3	6.0
PTMC/PLA/10%HA	307.1	335.4	347.9	378.0	5.8	5.3
PTMC/PLA/15%HA	267.1	333.0	352.9	395.1	16.5	15.8
PTMC/PLA/20%HA	246.5	311.0	365.0	362.6	19.6	19.3
PTMC/PLA/25%HA	231.7	293.0	—	340.1	26.0	25.9

**Table 3 nanomaterials-11-03215-t003:** Compressive strength and modulus of PTMC/HA and PTMC/PLA/HA scaffolds.

	Compressive Modulus (MPa)	F_σ_ = 50% (N)
PTMC	1.54	80.64
PTMC/5%HA	4.61	220.96
PTMC/10%HA	5.05	254.2
PTMC/15%HA	6.44	296.41
PTMC/20%HA	6.75	307.43
PTMC/25%HA	8.07	325.89
PTMC/PLA	348.34	3057
PTMC/PLA/5%HA	409.87	3513
PTMC/PLA/10%HA	381.22	3236
PTMC/PLA/15%HA	328.94	2961
PTMC/PLA/20%HA	320.89	2932
PTMC/PLA/25%HA	182.48	2019

## Data Availability

The data that support the findings of this study are available from the corresponding author upon reasonable request.

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
