# Peer review of "Biodegradable 3D Printed Scaffolds of Modified Poly (Trimethylene Carbonate) Composite Materials with Poly (L-Lactic Acid) and Hydroxyapatite for Bone Regeneration"

_nanomaterials, 2021, doi:10.3390/nano11123215_

Round 1

Reviewer 1 Report

  1. In abstract line 28, use acronyms for chemicals, previous line already mentioned the full name
  2. Line 32, and 33 only acronyms are given
  3. In line 35, what you mean by “drug controlled release”?
  4. In line 34, mechanic, or mechanical?
  5. In line 134, specify more details on 3d printing,(filament or used pellets)
  6. In experimental section, in a table specify samples with composition (eg different HA %)
  7. FTIR, results, site references
  8. in mechanical properties, the sample details are not mentioned (scaffold strut thickness, pore size etc)
  9. in line 399, “Subsequently, compressive strength and modulus of PTMC/PLA/HA composites gradually decreased again whilst the content of HA continued to increase from 5% to 25%.” Any explanation?
  10. In abstract, nothing mentioned about PTMC/HA scaffold

Reviewer 2 Report

In the manuscript entitled “Biodegradable 3D Printed Scaffolds of Modified Poly(trimethylene carbonate) Composite Materials with Poly(L-lactic acid) and Hydroxyapatite for Bone Regeneration” the authors Honglei Kang, Jiangxu Dong, Zhiwei Liu, Fan Liu, Feng Li, Guoping Yan explored the potential synergic effect of poly(trimethylene carbonate), poly(L-lactic acid) and hydroxyapatite in the preparation of a new set of 3D printed scaffolds for bone regeneration.

The topic is very central and widely discussed in the literature, but the novelty of the study reported in the paper is a little bit lacking. Indeed, the potential of 3D printed polylactic acid-hydroxyapatite scaffolds to promote new bone regeneration is well known and this paper seems just another little variation on the theme.

In the paper the authors propose a comparative analysis of the structural/mechanical properties of two composites made up of poly(trimethylene carbonate)/hydroxyapatite and poly(trimethylene carbonate)/poly(L-lactic acid)/hydroxyapatite. The stability and biocompatibility of the 3D printed scaffolds has also been discussed as well as their ability to promote bone regeneration. However, the results of these studies have not been adequately summarized in the conclusion section, where no indication on the best performing biomaterial is reported.

Another critical issue concerns the data in Figure 7c. In the graph of the MTT assay on the vertical axis is reported the relative expression, but it is not clear what does it means. The MTT assay is a colorimetric assay and typically the data are reported as the % of the variation of the absorbance at 570 determined by UV-visible spectrophotometric measurements. The authors should better explain what is reported in the graph.

On these basis, I recommend the publication of this manuscript in Nanomaterials after major revisions.

Reviewer 3 Report

The manuscript “Biodegradable 3D Printed Scaffolds of Modified Poly(trimethylene carbonate) Composite Materials with Poly(L-lactic acid) and Hydroxyapatite for Bone Regeneration” deals with the production of composite biopolymeric scaffolds for bone tissue engineering. Several analyses were performed on the obtained materials, achieving intriguing results. However, the work requires some revisions, as follows:

- The manuscript has to be organized in the Journal template (see author guidelines).

- Abstract. Add quantitative results to this section.

- Introduction. The state of the art related to the production of scaffolds using biopolymers and mixture of biopolymers, to obtain the proper physico-chemical features, can be enlarged; for this purpose, see for instance this work: Baldino et al., Nanostructured chitosan–gelatin hybrid aerogels produced by supercritical gel drying, Polymer Engineering and Science 58 (2018)1494 - 1499; etc. Moreover, in the aim of the work, reported at the end of this section, the novelty of the present study is not clear.

- Results. The graphs reported in Figure 1 are not clearly visible. Enlarge the figure or separate the graphs in two figures. SEM images reported in Figure 2 do not show any porosity of the samples. This is a required morphological properties for tissue engineering applications, in order to favor cell migration and proliferation. Discuss this results. Moreover, a systematic comparison of the results obtained in this work with the ones of the previous literature can be useful to highlight the novelty of the present study.

- References are not in the Journal template.

Round 2

Reviewer 1 Report

authors reply to to query was appropriate and they changed modification in the manuscript accordigly

Author Response

Thank you for your kind comments and suggestion on the structure and language of our manuscript.

Reviewer 2 Report

I appreciated the revisions made by the authors.

Unfortunately, there was a little misunderstanding for what concerns the last issue on the MTT assay.

In particular, the suggested modification of the y-axis title has been made not in the first graph reported in Figure 7c, now Figure 8c, but in the graphs reported in Figure 6, now Figure 7.

So, first of all I ask to change the title of the y-axis of the MTT garph in Figure 8c. Moreover, if you choose to report the cell viability (%) in Figure 7, then the control bar should not to be shown and all the data should be referred as percentage with respect to the control experiments.   

Author Response

Unfortunately, there was a little misunderstanding for what concerns the last issue on the MTT assay.

In particular, the suggested modification of the y-axis title has been made not in the first graph reported in Figure 7c, now Figure 8c, but in the graphs reported in Figure 6, now Figure 7.

So, first of all, I ask to change the title of the y-axis of the MTT graph in Figure 8c. Moreover, if you choose to report the cell viability (%) in Figure 7, then the control bar should not be shown and all the data should be referred as percentage with respect to the control experiments.

Thanks for your kind reminder, the Cell proliferation assays of the osteoblast cells MC3T3-E1 have been changed to Figure 7 in this revised version, y-axis of the MTT graph Figure 8c and figure 7 were all chosen to change to cell viability. The control displayed in the data was an error and has been deleted, cell viability (%) has been calculated as percentage with respect to the control. Sorry for the mistake in this graph.

Thank you again for your kind comments and suggestion about our manuscript.

Reviewer 3 Report

The authors improved the manuscript. However, the Introduction paragraph can be furtherly improved adding other references related to the production of porous structures obtained from biopolymeric blends (as in the case of this work: "Nanostructured chitosan–gelatin hybrid aerogels produced by
supercritical gel drying, Polymer Engineering and Science 58 (2018)1494 -
1499"; etc...). Some typos are present.

Author Response

The authors improved the manuscript. However, the Introduction paragraph can be furtherly improved adding other references related to the production of porous structures obtained from biopolymeric blends (as in the case of this work: "Nanostructured chitosan–gelatin hybrid aerogels produced by supercritical gel drying, Polymer Engineering and Science 58 (2018)1494 -1499"; etc...). Some typos are present.

Thank you for your kind comments and suggestion about our manuscript. The reference was added in the introduction paragraph. And typos of the manuscript have been revised.